# The Transcription Factor NF-κB in Stem Cells and Development

**DOI:** 10.3390/cells10082042

**Published:** 2021-08-10

**Authors:** Christian Kaltschmidt, Johannes F. W. Greiner, Barbara Kaltschmidt

**Affiliations:** 1Department of Cell Biology, Bielefeld University, Universitätsstrasse 25, 33615 Bielefeld, Germany; C.kaltschmidt@uni-bielefeld.de (C.K.); Johannes.Greiner@uni-bielefeld.de (J.F.W.G.); 2Molecular Neurobiology, Bielefeld University, Universitätsstrasse 25, 33615 Bielefeld, Germany

**Keywords:** NF-κB, REL, development, adult stem cells, embryonic stem cells, differentiation

## Abstract

NF-κB (nuclear factor kappa B) belongs to a family of transcription factors known to regulate a broad range of processes such as immune cell function, proliferation and cancer, neuroprotection, and long-term memory. Upcoming fields of NF-κB research include its role in stem cells and developmental processes. In the present review, we discuss one role of NF-κB in development in *Drosophila*, *Xenopus*, mice, and humans in accordance with the concept of evo-devo (evolutionary developmental biology). REL domain-containing proteins of the NF-κB family are evolutionarily conserved among these species. In addition, we summarize cellular phenotypes such as defective B- and T-cell compartments related to genetic NF-κB defects detected among different species. While NF-κB proteins are present in nearly all differentiated cell types, mouse and human embryonic stem cells do not contain NF-κB proteins, potentially due to miRNA-dependent inhibition. However, the mesodermal and neuroectodermal differentiation of mouse and human embryonic stem cells is hampered upon the repression of NF-κB. We further discuss NF-κB as a crucial regulator of differentiation in adult stem cells such as neural crest-derived and mesenchymal stem cells. In particular, c-REL seems to be important for neuronal differentiation and the neuroprotection of human adult stem cells, while RELA plays a crucial role in osteogenic and mesodermal differentiation.

## 1. Introduction to Canonical and Non-Canonical NF-kB Signaling

NF-κB (nuclear factor kappa B) was discovered in the laboratory of the Nobel Prize laureate David Baltimore as a latent transcription factor with inducible binding activity [1]. Later, Baeuerle and Baltimore described an inhibitory protein family named inhibitor κB (IκB), which could interact with NF-κB in the cytoplasm and was responsible for the latent form [2]. Molecular cloning revealed that the NF-κB family consisted of five DNA-binding members in humans: two without transactivation domains, NFKB1 (p50) and NFKB2 (p52); three with transactivation domains, REL (c-REL), RELA (p65), and RELB (RELB). In the canonical signaling pathway, NF-κB is formed by a heteromeric DNA-binding dimer, e.g., RELA (p65) or NFKB1 (p50), which is in its cytoplasmic (latent) form complexed with inhibitor of nuclear factor κB (IκB) [3]. This results in the allosteric induction of the closed non-DNA-binding conformation. IκB binding to the heterodimer RELA/p50 induces an alpha-helical conformation of the nuclear translocation signal sequence (NLS) on RELA, inhibiting the interaction with the nuclear import machinery [4]. For nuclear import, a disordered NLS sequence is important. IκB itself can translocate to the nucleus and might interact with DNA-bound NF-κB to induce nuclear export [5]. In its latent state, the RELA/p50 heterodimer exists in a stable cytosolic complex with a member of the IκB family (e.g., NFKBIA) (Figure 1, canonical pathway); see [6] for a comprehensive review. There are more than 300 proteins with IκB activity [7]. Extracellular stimuli for NF-κB include bacterial and viral products, inflammatory cytokines, reactive oxygen species, ultraviolet light, and even neurotransmitters such as glutamate [8,9]. For instance, inflammatory cytokines such as TNFα bind to TNFR1 inducing receptor trimerisation (Figure 1).

They activate a kinase complex (IKK, IκB kinase; see Figure 1) containing two related catalytic subunits, IKK-1 (or -α) and IKK-2 (or -β) [10]. An additional component, NEMO (NF-κB essential modulator or IKKγ), with no enzymatic activity has been cloned through the genetic complementation of an NF-κB activation-defective cell line [11]. Biochemical purification identified IKK-γ as an essential regulatory subunit of the IκB kinase complex [12]. The activated IKK complex catalyzes the sequence-specific phosphorylation (serine 32 and 36) and ubiquitination of the heterodimer-associated IκBα (Figure 1) [6,13]. This leads to the rapid degradation of IκBα through ubiquitin-mediated proteasomal degradation [14]. The removal of IκBα activates the NF-κB NLSs, facilitating NF-κB in binding to importins, resulting in its rapid translocation into the nucleus [15]. Within the nucleus, NF-κB binds to target gene promoters in a sequence-specific manner and activates gene transcription (see T. Gilmore’s webpage https://www.bu.edu/nf-kb/gene-resources/target-genes/ (accessed on 23 July 2021)).

Alongside canonical signaling, the binding of ligands (such as lymphotoxin-β) to the tumor necrosis factor (TNF) receptor superfamily member lymphotoxin-β receptor (LTβR) or the receptor activator for NF-κB (RANK) initiates the non-canonical NF-κB signaling pathway. Here, the NF-κB-inducing kinase (NIK)-mediated phosphorylation of IKK1, in turn, leads to the phosphorylation of p100, the precursor form of p52. The IKK1-dependent phosphorylation of p100 induces the processing of p100 to p52 [16], in turn, enabling the translocation of p52/RELB into the nucleus and the initiation of target gene transcription (Figure 1). Canonical and/or non-canonical NF-κB signaling is commonly known to mediate a broad range of processes such as proliferation, inflammation, and memory, but also autoimmune and inflammatory diseases as well as cancer (see [9,17,18,19,20] for a detailed review). In this vein, one role of NF-κB in neurological diseases was suggested as early as 1993 [21]. Extending the present literature, we summarize and discuss, in the following sections, the current knowledge regarding the role of NF-κB in the development of several species including *Drosophila*, *Xenopus*, mice, and humans, as well as its functions in mammalian stem cell biology.

## 2. NF-κB in Development

In the present review, we shed light on the universal concepts of NF-κB signaling (or that of its homologs) conserved during evolution by utilizing the evo-devo concept (evolutionary developmental biology, as reviewed in [22]). Recent bioinformatic analysis shed light on one of the evolutionary oldest proteins sharing a REL homology in the air-breathing freshwater snail *Biomphalaria glabrata* (reviewed in [23]). Interestingly, NF-κB-like proteins with a REL homology domain were even found in Cnidaria such as the sea anemone *Nematostella vectensis* [24] or the hydra *Hydractinia symbiolongicarpus* [25]. However, Cnidaria commonly show separate proteins harboring homologies to either an inhibitory ankyrin repeat domain or a REL domain, which are suggested to arose from gene-splitting events [24,26] (reviewed in [23]). Notably, the NF-κB pathway seems to have been lost to a great extent in nematodes such as *Caenorhabditis elegans*, which do not comprise proteins homologues to NF-κB (reviewed in [23]). On the contrary, mutations in the IκB homologs NFKI-1 and IKB-1 were found to be associated with developmental defects in *C. elegans*, proposing a role of NFKI-1 and IKB-1 in chromatin regulation [27].

Going further in evolution, a protein with REL homology was historically the first discovery by the team of Christiane Nüsslein-Volhard in the fly *Drosophila melanogaster* as Dorsal (Dl), the regulator of dorsoventral pattern formation [28]. For their discoveries concerning the transcription factor cascades directing development, Christiane Nüsslein-Volhard was awarded the Nobel Prize in 1995 for Physiology or Medicine. Drosophila has two NF-κB family members with DNA-binding domains: transactivating Dorsal (Dl) and DIF (or Dorsal-related immunity factor), with a typical REL homology domain, which is 45% identical to that of the mammalian counterparts c-REL, RELA, and RELB. This high homology is surprisingly conserved for these proteins despite an evolutionary distance of more than 800 million years. Both proteins are retained in the cytoplasm through binding to an inhibitor protein, Cactus, which is homologous to mammalian IκBs [29]. After the fertilization of the Drosophila egg (Figure 2A), a gradient of Dl in the nuclei of the syncytium (Figure 2B, Dl colored in blue) is formed by positive regulators such as Spätzle on the ventral side. This gradient is shaped by extracellular repressors of Dl signaling such as Decapentaplegic (Dpp; Figure 2C, Dpp colored in red). Short gastrulation protein (Sog) antagonizes Dpp signaling at the ventral side (Figure 2C). The dorsoventral axis in Drosophila results in the formation of a gradient of the NF-κB RELA homolog Dl. Two to four hours after fertilization, the Dl gradient regulates various target genes. The highest levels of Dl activate target genes inducing mesoderm formation, whereas intermediate levels of Dl induce epidermal growth factor (EGF) signaling, which is important for neurogenic ectoderm formation. Interestingly, not only is the nervous system in vertebrates turned from ventral to dorsal but, also, its regulators such as bone morphogenetic factor-4 (BMP-4) and its antagonist Chordin (a homolog of fly Sog, colored in light blue in Figure 2D). This interesting developmental cue was detailed by investigations of Edward De Robertis [30]. Similarly, NF-κB in mice was constitutively active when transforming growth factor-beta 2 (TGF-ß) was not expressed in the developing cerebellum [31]. In this vein, we could show that TGF-ß is essential for the proliferation of cerebellar granular cell precursors in cerebellar organotypic slice cultures [32].

In vertebrate evolution, AmphiREL [33], a Rel-like gene identified in *Brachiastoma belcheri*, seems to be the ancestry protein evolved to the other vertebrate family members RELB and RELA by whole-genome duplication events in jawed and jawless vertebrates, with three duplications being present in zebrafish [34]. Following the evo-devo concept, developmental pathways in the lower developed fly may also suggest a role of NF-κB in frog development. Indeed, a homolog of REL was cloned from *Xenopus laevis* and named Xrel3, which is expressed in the late neurula stage in the developing brain [35]. Xrel3 cannot induce head formation on its own. However, Xrel3 can initiate tumor formation at the late neurula stage. Xrel3 has a C-terminal transactivation domain similar to RELA [36]. A C-terminal truncated version of Xrel3 acts as a trans-dominant negative protein. The injection of the truncated form of the Xrel3 mRNA at the dorsal site of *Xenopus embryos* led to massive head defects up to headless embryos (anencephaly). These data could be reproduced with a trans-dominant version of IκB, resulting in defects in axis formation, with no head formation at high amounts [37]. Furthermore, the same authors identified Wnt/beta-catenin signaling as an upstream activator of Xrel3.

A function of NF-κB in the notochord was detected during fish development. The notochord is an essential feature of chordates. This structure can send key signals for patterning during the development of tissues surrounding the notochord. The overexpression of a dominant-negative form of the murine IκBα in zebrafish resulted in defects of notochord development, generating no-tail embryos [38]. Furthermore, NF-κB is essential for the production of hematopoietic stem cells in zebrafish [39]. In contrast to frog mutants, the knockout (KO) of NF-κB subunits in mice did not generate gross anatomical defects. All the reported defects are summarized in Table 1.

RELA knockout resulted in liver failure due to TNFα-mediated apoptosis [50]. The liver phenotype of RELA KO mice could be rescued by a double KO of TNFR1 and RELA [47]. Thus, we conclude that TNFα is a major driver of liver apoptosis in RELA-KO mice. C-REL knockout and p50 knockout do not produce any gross anatomical phenotype, but RELB knockouts lead to defects in the immune system (for a review, see [40]). In particular, p50 KO creates cellular defects resulting in decreased survival, defects in the immune system such as in T-cell development, and defects that create hearing loss [51]. However, the tissue-specific expression of trans-dominant IκB leads to defects in neurogenesis within the dentate gyrus of the hippocampus [52]. The relief of NF-κB repression leads to the regeneration of the adult hippocampus. A loss of function induced by the expression of dominant-negative IKK2 in the forebrain neurons [53] resulted in strongly reduced Igf2 expression in neurons and reduced spine formation on the neurons of the hippocampal region CA1. Furthermore, the postsynaptic densities changed, presenting a patchier appearance. Surprisingly, the overexpression of the trans-dominant gain of function form of IKK (CamK2a-tTA × luciferase-(tetO)7-IKK2-CA, called IKK2^nCA^) also leads to defects within the dentate gyrus, with reduced BDNF expression and high latency in the “Morris water maze” behavioral test in nine-month-old mice [54].

Mutants of the NF-κB subunits REL and RELA are only rarely detected in humans (own analysis using OMIM (*Online Mendelian Inheritance in Man*). For RELA, haploinsufficiency was described [55], where biallelic expression might be necessary for mucosal integrity. While the deletion of RELA in HEK 293 cells [45] or monoallelic inactivation in human fibroblasts [55] has no effect on TNF-α-mediated apoptosis, the oral mucosa of afflicted patients developed ulcers. Taken together, RELA^−/+^ might result in the impairment of stromal and epithelial cell recovery in response to mucosal injury. Furthermore, a mutant allele for RELB detected in patient lymphocytes showed a loss of function of the RELB protein. Patient lymphocytes were found to have a homozygous mutation that created a premature stop codon, which blocked the translation of the last three exons. RELB deficiency was identified in patients with repeated infections, failure to thrive and autoimmunity [49]. The CD4^+^:CD8^+^ ratio was increased compared to that in the controls. In vitro studies showed impaired T-cell proliferative responses to multiple antigens as well as an impaired ability to produce specific immunoglobulins, indicating an additional effect on humoral immunity. RELB deficiency could be cured by hematopoietic stem-cell transfer [56].

## 3. Components of NF-κB Signaling Affected by Mutations in Human Patients and Cellular Models

The medical relevance of murine studies might be limited by, for example, certain differences such as the greater ease of transforming murine cells or differences between mouse and human immunology [57].

Genetic screens identified a truncated form of NEMO in incontinentia pigmenti patients, which is defective in NF-κB activation [58]. Patients suffer from various conditions such as abnormalities in the developing retinal vessels; missing or deformed teeth; and, rarely, convulsive or motor disorders or mental retardation. In accordance with the localization of NEMO on the X chromosome [59,60], male individuals and male mice with NEMO deficiency die from TNF-mediated liver apoptosis, while females develop normally but, soon after birth, exhibit patchy skin lesions with massive granulocyte infiltration. Furthermore, a gain-of-function mutation of IκBα with a missense mutation of serine 32 was reported [61], with a phenotype similar to that of the NEMO mutation: dominant anhidrotic ectodermal dysplasia and T-cell immunodeficiency. Furthermore, IκBα mutations such as truncations were detected in Morbus Hodgkin patients [62], leading to constitutive NF-κB activity [63]. Likewise, Bredel and coworkers demonstrated that deletion of NFKBIA is associated with short survival of patients suffering from glioblastoma. By comparing 790 human glioblastomas, the authors reported NFKBIA deletion in glioblastoma to result in clinical outcomes comparable to EGFR amplification. They report heterozygous deletion of NFKBIA in up to 28% of glioblastomas in their study and in 22% of cancer stem-like cells. There seems to be a mutual exclusion of either EGFR amplification or NFKBIA deletion. Interestingly, forced expression of NFKBIA by retroviral transduction of established glioblastoma cell lines inhibited signs of malignant transformation such as growth, migration, and colony formation [64]. In addition, RELA mutations are now within the consensus panel of diagnostic markers for glioblastoma [65], and RELB overexpression was reported to be correlated with the grade of the glioma [66].

An IKK2 deficiency, which is lethal in mice, leads to non-lethal defects in adaptive and innate immunity in humans [67]. A search for SNPs in SNPedia revealed some association of the REL locus with rheumatoid arthritis and Crohn’s disease. For RELA, there is a single report of a missense mutation (p.Ile110Thr), linked to childhood-onset schizophrenia. For RELB, a pathogenic missense mutation was reported (p.Tyr397Ter). Furthermore, NFKB2 genetic polymorphisms were associated with lymph node infiltration in colorectal cancer. On a single-cell level, the CRISPR/Cas9-mediated knockout of both RELA alleles in human HeLa cells resulted in a profound defect in the cell cycle [45]. Specifically, either the cell cycle was completely blocked, or prometaphase was delayed. Several target genes such as MYC and intercellular adhesion molecule 1 (ICAM) were reduced in expression by more than 50%. REL^−/−^ cells were more resistant to cisplatin. In the human kidney cells HEK293, the double knockout of IKK1 and IKK2 blunted NF-κB activation and rendered cells 10 times more sensitive to TNFα-induced cell death [68]. RELA knockout strongly reduced the TNFα-mediated induction of target genes such as IκB-alpha, TNF-α, and A20. TNFα-mediated cell death was not influenced in RELA-knockout cells, in contrast to in mice with genomic deletion, which die from TNFα-mediated apoptosis of the liver (see Table 1) [50]. We conclude that defects of REL in human cells profoundly affected the cell cycle, which might explain why no vital human mutant with c-REL defects has been observed to date (own literature search in 2021).

## 4. NF-κB Signaling in Embryonal Stem Cells

Human embryonic stem cells (ESCs) can be derived from the inner cell masses of human blastocysts [69]. ESCs are pluripotent and can form undifferentiated embryonic tumors—teratomas—in immunocompromised mice [69]. ESCs spontaneously form embryoid bodies comprising the three germ layers: ectoderm, mesoderm, and endoderm [70]. The Nobel Prize laureate Yamanaka discovered that mouse and human mature cells could be reprogrammed to become pluripotent ESCs (iPS) by the forced expression of four transcription factors: Oct4, Klf4, Sox2, and MYC [71].

The expression of NF-κB NFKB1 and RELA at the mRNA and protein levels in human ESCs was analyzed [72]. It was observed that NFKB1 and RELA were expressed at extremely low levels in these cells at the mRNA level. The NFKB1 and RELA proteins were not detected in extracts from human ESCs.

Similarly, NF-κB activation might interfere with the generation of iPS. Genetic and pharmacological NF-κB inhibition significantly increased the efficacy with which fibroblasts were reprogrammed to iPS [73] (the paper was later retracted due to problems with the Western blotting). Furthermore, the differentiation of ESCs with retinoic acid (RA; 10^−5^ molar) for 10 to 15 days greatly increased the mRNA levels of p65 and p50. After six days of RA treatment, nuclear localization of p65 was detected. Similarly, in mouse ESCs, no p65 protein was detected, although the mRNA of p65 was already present in undifferentiated ESCs [74]. When mouse ESCs were differentiated with RA, the NF-κB p65 protein was highly upregulated. The repression of the translation of p65 mRNA in undifferentiated mouse ESCs was shown to be mediated by the expression of the miR-290 cluster (see below for further discussion regarding the role of miRs in regulating NF-κB in ESCs).

In human ESCs, the knockout of RELA did not interfere with cell death and teratoma formation, but defects in differentiation were observed. There were 20% more adipocytes in RELA^−/−^ cells after differentiation. In addition, osteogenesis and chondrogenesis were defective [48]. Strikingly, after seven passages, there were no MSC progeny among differentiated RELA^−/−^ cells. By contrast, the number of vascular endothelial cells increased by about 50% after differentiation, while vasculogenesis was impaired and cells had become hypersensitive to TNFα-induced apoptosis. To a lesser extent, TNFα-induced apoptosis was detected in RELA^−/−^ MSCs (see Section 6). In this vein, nuclear RELA was detected in human atherosclerotic lesions of vascular cells [75], potentially activating a cytoprotective gene expression program in human cells.

When the level of NF-κB is reduced by manipulating the expression of IKK, two phenotypes arise after RA-mediated differentiation:Differentiation with high levels of NF-κB gives rise to progeny of mesodermal origin (SMA positive);Differentiation with intermediate levels of NF-κB results in neuro-ectodermal progeny (Tuj-positive) at the expanse of the mesoderm [76].

Furthermore, the NF-κB-activity in hESCs was measured using an NF-κB-driven GFP reporter system during neuronal differentiation [77]. The authors showed that NF-κB-driven reporter gene activity was active only after differentiation to the stage of self-renewing neural progenitor cells (NPCs). Interestingly, NPCs with low NF-κB activity were SOX2^+^, Lin28^+^, and Nestin^+^, whereas NPCs with high NF-κB reporter gene activity were SOX2^−^, Lin28^−^, and Nestin^+^. Furthermore, NF-κB^high^ NPCs gave rise to more than 60% of neurons with β-3-tubulin expression, whereas these represented about 1% in differentiated NPCs with low NF-κB-activity. Of note, LIN28, a marker of undifferentiated ESCs together with Sox2, could repress NF-κB in ESCs (reviewed by [78]).

As discussed above, the miR-290 cluster was reported to repress the translation of p65 mRNA in undifferentiated mouse ESCs [74]. Another set of miRNAs—miR-371, miR-372, miR-373*, and miR-373—on chromosome 19 is the human homolog of mouse miR-290, miR-291-s, miR-291-as, miR-292-s, miR-292-as, miR-293, miR-294, and miR-295 expressed in mouse ESCs [79]. Let-7a is also expressed in human ESCs [80]. Since miR-290 repressed NF-kB activity in mouse ESCs, we suggest that homologous miRNAs might act in the same manner in humans.

## 5. Regulatory Roles of NF-κB in Driving Differentiation of Neural Crest-Derived Stem Cells

Adult stem cells with an embryonic origin from the neural crest (NC) can be found in various niches of the human organism including the skin and its hair follicles [81,82], the bone marrow [83], the heart [84,85], the dental pulp [86], or the respiratory and olfactory epithelium of the nasal cavity [87,88,89]. Neural crest-derived stem cells (NCSCs) are characterized by an extraordinary high potential for differentiation, particularly into mesodermal and ectodermal cell types, making them promising candidates for pharmacological research and regenerative medicine [90,91,92,93,94,95,96] (reviewed in [97]). On a mechanistic level, transcription factors such as Slug, Sox10, and Twist commonly associated with NC development and epithelial-to-mesenchymal transition (EMT) are increasingly suggested to regulate the stemness of adult stem cells, particularly NCSCs [98,99] (reviewed in [100]). Since NF-κB is known as a potential regulator of these EMT/NC drivers [101,102,103], we summarize, in the present section, the current knowledge regarding the role of NF-κB-signaling in NCSCs and their differentiation into the osteogenic and neuronal lineages.

The exposure of adult neural crest-derived stem cells to common activators of NF-κB signaling such as lipopolysaccharides (LPS) or polyinosinic:polycytidylic acid (Poly(I:C)) is broadly described to stimulate NF-κB activity. In this regard, we describe the activation of NF-κB RELA by LPS and Poly(I:C) as mimicking bacterial and viral infections in inferior turbinate stem cells (ITSCs) of the human nasal cavity [93]. Accordingly, He and coworkers reported LPS to induce the expression of IL-8 in an NF-κB RELA-dependent manner in human dental pulp stem cells (DPSCs) [104]. Interestingly, the activation of NF-κB was accompanied by increased phosphorylation of p38-MAPK [104], which we very recently identified as a crucial regulator of cell proliferation and migration in human NC-derived adult cardiac stem cells [85,105].

In terms of the differentiation of NCSCs into osteogenic cell types, Feng and coworkers demonstrated that TNF-α triggered the osteogenic differentiation of human DPSCs via the activation of NF-κB. The authors particularly observed elevated translocation of NF-κB RELA into the nuclei of DPSCs, resulting in increased mineralization as well as elevated expression of bone morphogenetic protein 2 (BMP2), runt-related transcription factor 2 (RUNX2), and collagen type I upon treatment with TNFα during osteogenic differentiation. Accordingly, the osteogenic differentiation of DPSCs was blocked by the application of the NF-κB inhibitor pyrrolidine dithiocarbamate (PDTC) [106]. On the contrary, DPSCs derived from estrogen-deficient rats revealed a downregulation of osteogenic differentiation potential, accompanied by elevated NF-κB activity [107], which is in line with the reduced potential for differentiation into osteogenic derivates observed in human stem cells from female individuals [108,109] (reviewed in [110]). The application of LPS to NC-derived periodontal ligament stem cells (PDLSCs) was further shown to decrease their osteogenic differentiation potential via toll-like receptor-4 (TLR-4)-dependent activation of NF-κB in vitro. The inhibition of TLR4 or NF-κB also prevented bone loss in a rat model of periodontitis [111]. In this vein, Chen and colleagues demonstrated defective osteogenic differentiation capability for PDLSCs isolated from periodontitis patients, which could be rescued by the application of BAY 11-7082—an inhibitor of IκBα phosphorylation—or PDTC. Interestingly, the authors observed NF-κB to compete with Wnt signaling in PDLSCs from periodontitis patients, although the exact molecular mechanisms still remain unknown [112]. NF-κB thus seems to play highly diverse roles in the regulation of the osteogenic differentiation of NCSCs, influenced by the niche or stem cell type, and there are differences between rodents and humans as well as in terms of the sex of the individual/animal. While benefitting the osteogenic differentiation of human DPSCs [106], NF-κB activation in estrogen-deficient rat DPSCs or human PDLSCs was directly linked to defects in their osteogenic differentiation potential [107,111,112]. The findings discussed here, therefore, emphasize the stem cell type/niche, organism, and sex of the organism as crucial parameters for assessing the role of NF-κB in the osteogenic differentiation of NCSCs.

In terms of the role of NF-κB in guiding the neuronal differentiation of NCSCs, we very recently observed NF-κB c-REL to be highly active during the differentiation of NC-derived ITSCs into glutamatergic neurons (see Table 1). In particular, differentiating human ITSCs revealed a peak of c-REL activity during a period from the second to the fifth day of guided neuronal differentiation. Notably, the inhibition of c-REL by the application of pentoxifylline (PTXF) resulted in a complete fate shift towards OLIG2^+^/O4^+^ oligodendrocytes, accompanied by decreased neuronal survival (see Table 1). Predifferentiated ITSCs treated with PTXF successfully integrated into an ex vivo oxidative stress-mediated demyelination model of mouse organotypic cerebellar slices, followed by differentiation into MBP^+^ oligodendrocytes [46]. Alongside the role of NF-κB in directly regulating the neuronal differentiation of human NCSCs, we identified NF-κB RELA as a key player in the neuroprotection of ITSC-derived glutamatergic neurons against oxidative stress. Notably, although the activation of NF-κB RELA by TNFα resulted in the protection of neurons differentiated from ITSCs of both sexes, we found the protective gene expression program beneath to be sexually dimorphic [113] (reviewed in [110]). In summary, NF-κB activity is vital for guiding the fate decisions of NCSCs between neurogenesis and oligodendrogenesis as well as in terms of neuroprotection, although these roles of NF-κB are highly subunit-specific.

## 6. NF-κB as a Mediator of Differentiation and Migration of Adult Mesenchymal Stem Cells

Alongside its role in neural crest-derived stem cells, NF-κB is controversially discussed as a driver of differentiation for mesenchymal stem cells. In this regard, the chondrogenic and myogenic differentiation of MSCs was reported to be negatively regulated by NF-κB [114,115]. In particular, Shakibaei and colleagues reported the inhibition of NF-κB by curcumin in IL1-treated MSCs to result in increased production of type II collagen and cartilage-specific proteoglycans [116]. In this vein, the authors likewise showed the curcumin-dependent protection of human chondrocytes against apoptosis [117]. Hu and colleagues demonstrated a magnesium-dependent inhibition of NF-κB in macrophages, while a macrophage-conditioned medium with magnesium treatment enhanced the chondrogenic differentiation of human bone marrow MSCs [118]. In terms of myogenic differentiation, NF-κB upregulation was shown to inhibit the myogenesis of MSCs in a cyclinD1/CDK4-dependent manner, which, in turn, inhibited the key myogenic transcription factor MyoD1 [119]. Along this line, Proto and colleagues revealed that the pharmacological or genetic inhibition of NF-κB resulted in the elevated myogenic differentiation of muscle-derived progenitor cells [120], which belonged to the mesenchymal stem cell lineage [121,122]. The observations strongly suggest that the inhibition of NF-κB leads to a microenvironment antagonizing pro-inflammatory conditions, which, in turn, facilitates the chondrogenic and myogenic differentiation of MSCs [116,123].

On the contrary, Hess and coworkers demonstrated an increase in the osteogenic differentiation of hMSCs after stimulation with TNFα or the expression of constitutively active IKK2 [124]. In particular, the activation of NF-κB in hMSCs resulted in increased expression of BMP-2, Runx2, and Osterix, as well as elevated mineralization. On the contrary, the authors observed no decrease in the osteogenic differentiation capability of hMSCs after genetically blocking NF-κB signaling, suggesting that NF-κB supported osteogenic differentiation without being an absolutely necessary prerequisite [124]. In line with these observations, Lombardo and colleagues showed that the exposure of hMSCs derived from adipose tissue to TLR-4 ligands increased the generation of osteogenic derivates, while proliferation and adipogenic differentiation remained unaffected [125]. The pretreatment of human osteoblasts with TNFα for 24 h was further demonstrated to direct the osteogenic differentiation of cocultured MSCs in vitro, leading to the proposal of a TNFα-mediated paracrine loop driving MSC-lineage commitment [126]. Contrary to the findings of Hess and Lombardo, Chang and colleagues reported that TNFα and IL-17 impaired the osteogenic differentiation of hMSCs via the activation of IKK and NF-κB, in turn resulting in the ubiquitination and degradation of β-catenin. Accordingly, the application of a small-molecule inhibitor of IKK (IKKVI) increased the osteogenic differentiation of hMSCs in vitro, as well as MSC-mediated craniofacial bone repair in vivo [127]. We, therefore, conclude that NF-κB may play different roles in driving or preventing the osteogenic differentiation of MSCs in a context-dependent manner, which is in line with its dual roles in regulating the osteogenic differentiation of NCSCs (see above).

In addition to mediating differentiation, NF-κB was demonstrated to induce the migration and proliferation of human MSCs in response to pro-inflammatory cytokines. In this regard, Carrero and colleagues showed interleukin-1ß (IL-1ß) to induce global gene expression related to cell survival and migration, while the inhibition of NF-κB via the knockdown of IKKβ impaired the migration and adhesion of MSCs [128]. In this vein, the stimulation of hMSCs with TNFα strongly augmented their migration and proliferation behavior in an IKK2–NF-κB-dependent manner via the upregulation of cyclin D1 gene expression [129]. The activation of NF-κB in MSCs by preconditioning with diazoxide, an activator of ATP-sensitive potassium ion channels, increased the survival of MSCs subjected to oxidative stress in vitro. The transplantation of preconditioned MSCs in a female rat model of acute myocardial infarction resulted in elevated cell survival, angiomyogenesis, and functional improvements compared to non-preconditioned MSCs or preconditioned MSCs pretreated with an NF-κB inhibitor [130]. These promising observations strongly suggest a beneficial role of NF-κB in regulating cell survival and the migration and adhesion of MSCs, in turn facilitating recruitment to the site of injury in regenerative contexts [129,130].

## 7. Discussion and Concluding Remarks

In summary, the present review discusses current knowledge regarding the complex roles of NF-κB-signaling in development as well as in embryonic and adult stem cells. Here, we used an evo-devo concept to detect signaling modules conserved over a wide evolutionary distance, starting from flies (*Drosophila*) and progressing to vertebrates: frogs (*Xenopus*) and humans. In addition, we included cellular models as the industry standard for modern high-throughput screening (HTS) systems relevant for the design of cell-based luciferase reporter gene assays [131]. Here, we review which phenotypes are evolutionarily conserved and which effects might be detected in cellular models. We conclude that, despite broad evolutionary distances, REL domain-containing proteins of the NF-κB family are evolutionarily conserved among highly different species including *Drosophila*, *Xenopus*, mice, and humans. In Drosophila, a gradient of Dorsal, the fly homolog of vertebrate NF-κB, mediates dorsoventral axis development, with low Dorsal levels being a prerequisite for neuroectodermal development [28,43]. In vertebrates, this gradient orientation is turned from ventral to dorsal (see Figure 2). In contrast to the nervous system, which is well-established in flies, the immune system is a recent invention evolved by whole-genome duplication events in jawed and jawless vertebrates, with three duplications present in zebrafish [34]. The older evolutionary function of NF-κB seems to be the regulation of neuroectodermal development. The role of NF-κB in regulating the immune system was shaped during fish evolution. The NF-κB homolog Xrel3 in frogs was further reported to regulate the induction of the head [37]. From these effects on neuroectodermal development in flies and frogs, additional cellular defects in mice were reported. While NFKB1 (p50) ensures the proper development of the T-cell compartment in mice [40], in humans, the B-cell compartment [42] is regulated by NFKB1. A KO of murine RELA resulted in embryonal lethality at E15-16 by the TNFα-mediated apoptosis of hepatocytes, while heterozygous RELA deletion is known to be associated with oral ulcers in patients (see Table 1). We, therefore, conclude that, despite its broad evolutionary conservation, the role of NF-κB and its homologs during development extends from solely mediating dorsal–ventral patterning (fly) toward complex immune cell functionality and stem cell behavior (mammals). In this regard, research found the differentiation of ESCs with high levels of NF-κB to result in progeny of mesodermal origin, while intermediate levels of NF-κB led to the presence of neuroectodermal rather than mesodermal progeny [76]. These findings demonstrate the crucial influence of NF-κB on the differentiation of mammalian ESCs and are also in line with the observation of low Dorsal levels being a prerequisite for neuroectodermal development in the fly. Notably, the mouse miRNA cluster 290 [74] and its human homolog miR-371/372/373 were found to inhibit the translation of NF-κB mRNA in embryonic stem cells, in turn resulting in the absence of NF-κB proteins [76]. We, therefore, suggest that NF-κB downregulation at the protein level is a prerequisite for maintaining the pluripotent state.

Aside from its role in embryonic stem cells, NF-κB plays a crucial role in adult stem cells with an embryonic origin from the neural crest. In particular, EMT/NC-related transcription factors suggested to regulate NCSC stemness are known to be transcriptionally driven by NF-κB [100]. Our own recent observations further showed the pivotal role of NF-κB c-REL in the fate choice of NCSCs between neurogenesis and oligodendrogenesis [46]. RELA was further observed to mediate the neuroprotection of NCSC-derived neurons [113], emphasizing the subunit-specificity of NF-κB signaling in the neuronal differentiation of NCSCs. For MSCs, NF-κB activity was shown to inhibit myogenic and chondrogenic differentiation, potentially by promoting a pro-inflammatory environment [116,123]. On the contrary, we further conclude that NF-κB may play a dual role in the osteogenic differentiation of MSCs [124,125,127] and NCSCs [106,107,111,112], namely, by promoting or preventing specialization into osteoblasts. These multiple roles of NF-κB in regulating the differentiation of ASCs emphasize the importance of considering parameters such as the stem cell type/niche, organism, and sex of the organism for assessing NF-κB signaling in stem cells in future studies. Of note, model-specific limitations might affect the applicability of the results to the human system. As an example, hepatocyte phenotypes (apoptosis) and liver cancer were detected in mice with loss of function or gain of function of NF-κB activity, but these have not been detected in humans to date (see Table 1). More specifically, an example of a loss of function mutation is a KO of Nemo causing hepatocellular carcinoma in mice with increased hepatocyte apoptosis in early stages [132]. Nemo/IKKß-deletion in hepatocytes renders mice sensitive to induction of hepatocarcinomas by diethylnitrosamine [133]. However, human IKBKG (NEMO) has no prognostic value in cancer diagnostics (the human protein atlas, www.proteinatlas.org (accessed on 23 July 2021)). As a prognostic marker, a high expression of human RELA is unfavorable in renal cancer and liver cancer, although the 5-year survival rate (50%) of liver cancer patients with high RELA expression differs only slightly compared to patients showing low expression of RELA (38% 5-year survival rate, the human protein atlas, www.proteinatlas.org (accessed on 23 July 2021)). In contrast, high RELB expression is favorable in breast cancer (84% 5 year survival with high expression and 73% with low RELB expression), but unfavorable in renal cancer (63% 5 year survival with high expression and 81% with low RELB expression). Immunolocalization of RELB in glandular tissue of renal adenocarcinomas is sometimes 75% nuclear with moderate staining intensity. Taken together, the here discussed findings emphasize the need to use human cellular models as it is already industry standard in cancer research as in the case of tumor organoids [134].

## Figures and Tables

**Figure 1 cells-10-02042-f001:**
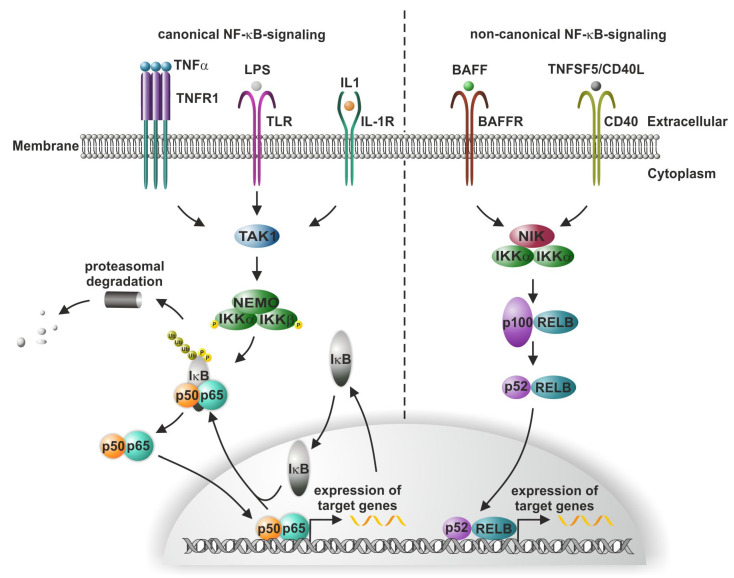
Schematic view of canonical and non-canoncial NF-κB signaling in mammals. In canonical signaling, NF-κB exists in two forms: a latent form complexed with IκB retained within the cytoplasm, and activated NF-κB without inhibitory subunits, binding to DNA in the nucleus. Various extracellular stimuli activate NF-κB, such as bacterial and viral products, inflammatory cytokines, reactive oxygen species, ultraviolet light, and even neurotransmitters. Intracellularly, a kinase complex (IKK, IκB kinase) is activated, in turn, resulting in phosphorylation and linear ubiquitination of the heterodimer-associated IκB. Rapid degradation of IκB through ubiquitin-mediated proteasomal degradation is followed by nuclear import of NF-κB and binding to target gene promoters such as IκB. Later, IκB can enter the nucleus and terminate the expression of NF-κB-target genes. In non-canonical signaling, binding of ligands to respective receptors such as BAFFR leads to NF-κB-inducing kinase (NIK)-mediated phosphorylation of IKK1, which in turn results in processing of p100 to p52 and enables the translocation of p52/RELB into the nucleus and the initiation of target gene transcription. Expression patterns in human single cells measured by RNA sequencing differed considerably. Whereas RELA has low cell type-specificity, c-REL is predominantly expressed in monocytes and keratinocytes and RELB in monocytes and glandular cells (The Human Protein Atlas, www.proteinatlas.org (accessed on 23 July 2021)).

**Figure 2 cells-10-02042-f002:**
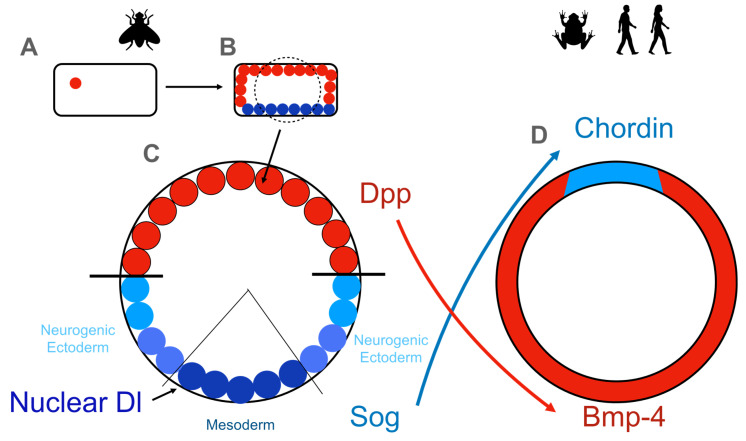
Schematic view of the fly Drosophila melanogaster’s development (in (**A**–**C**)), also shown in comparison to that of vertebrates such as the frog *Xenopus laevis* and humans (in (**D**)). A fertilized Drosophila egg (**A**) starts to divide and forms an embryo where the cytoplasm is shared between all nuclei (syncytium, (**B**)). At this stage of development, a dorsoventral gradient initiates the nuclear localization of Dorsal (Dl, blue), which is the fly homolog of vertebrate NF-κB. Dl is nuclear in the ventral region, as can be seen in the cross-section of the fly embryo depicted in (**C**). Nuclei without Dl are depicted in red, and a graded amount of nuclear Dorsal is depicted in shades of blue, with the maximal Dl concentration in dark blue. Extracellular regulators such as Decapentaplegic (Dpp) inhibit nuclear orsal, whereas Dpp is inhibited by its antagonist Sog (short gastrulation protein). Vertebrates do not have a ventral nerve cord but have a spinal cord; thus, it would not be entirely surprising if molecular regulators were also turned in their orientation from ventral to dorsal. BMP-4, a homolog of Dpp, is expressed in the ventral domain (**D**), whereas Chordin, a homolog of Sog, is expressed in the dorsal domain.

**Table 1 cells-10-02042-t001:** Cellular phenotypes related to genetic NF-κB defects detected among different species.

Subunit	Fly	Frog	Mouse (for Review, See [40])	Human
**P1005/P50**	No	Yes	Decreased survival, defect in the immune system, T-cell development defect; hearing loss	NFKB1 B-cell dysfunction [41]
**P100/p52**	No	Yes	Gastric hyperplasia, enhanced cytokine production by T-cells; poor antibody response	NFKB2 common variable immunodeficiency (CVID); antibody deficiency; impaired B-cell development [42]
**c-REL**	Dorsal for dorsoventral axis [28], high Dorsal: mesoderm, low Dorsal: neuroectoderm [43]	Xrel3 for axis formation, head induction [37]	Defects in the hematopoietic system; neuronal survival defect, late-onset (starting after 18 months); Parkinson’s-like disease [44]	REL KO results in defects in cell cycle [45]; necessary in neural crest cells for neuronal development and survival; inhibition results in fate switch into oligodendrocytes [46]
**RELA**	See above for c-REL/Dorsal	n.a.	RELA KO resulted in embryonal lethality at E15-16 by TNF-mediated apoptosis of hepatocytes; RELA^−/−^ TNFR1^−/−^ show spatial learning and synapse formation defects [47]	RELA KO resulted in defective vascular endothelial and mesoderm differentiation in iPS cells [48]; RELA^−/+^ resulted in oral ulcers in patients
**RELB**	See above for c-REL/Dorsal	n.a.	KO resulted in inflammation and hematopoietic abnormality	RELB KO resulted in increased nuclear RELA; hyperactivation of RELB by TCR activation [49]

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
