# Peer review of "The Transcription Factor NF-κB in Stem Cells and Development"

_cells, 2021, doi:10.3390/cells10082042_

Round 1

Reviewer 1 Report

This review focuses on the evolution of NF-κB signaling and its clinical implications. However, the review is not balanced since some sections describing the developmental defects associated with NF-κB mutations are clear and informative while others and misleading. In addition, the more general sections on NF-κB regulation, IκBs and IKKs are highly confusing and sometimes repetitive.

Multiple sentences and concepts in the review are obscure and the messages incomplete or biased, when the objectives of a review manuscript are to provide useful and understandable information to the readers. Only some examples:

“In this review, we discuss a role of NF-κB in development from Drosophila via Xenopus and mice to humans.” What does it mean “via xenopus and mice”? Are authors referring to a specific evolutionary axis? Then, they also talk about fish, is fish in the same axis?

“NF-κB is formed by a heteromeric DNA-binding dimer e.g. RelA (p65) and NFKB1 (p50) complexed with IκB”. I do not think that IκB is a subunit of the NF-κB complex. This could apply, at most, to the cytoplastic (inhibited) complex.

Extracellular stimuli of NF-κB include bacterial and viral products, inflammatory cytokines, reactive oxygen, ultra- 40 violet light and even neurotransmitters such as glutamate [8,9]. They activate a kinase complex (IKK, IκB kinase, see Fig. 1) specific for IκB”. Saying that IKK complex is specific for IκB is a statement that shouldn’t be done since several non-IkB targets for IKKs have been identified including FOXO3A, SMRT, SNAP-23, histone H3 or ATM.

“IKK complex catalyzes a sequence-specific phosphorylation (serine 32, 36) and ubiquitination of the heterodimer-associated IκB”. They are referring to IκBalpha since residues phosphorylated in other IκBs are different.

“Developmental pathways would suggest a role of frog NF-κB in development.” I do not understand the meaning of the sentence.

In the section “NF-κB in development” authors mention that “RelA knockout resulted in liver failure due to TNF-alpha-mediated apoptosis”, which to my view is part of the general role for NF-κB as inhibitor of apoptosis.

“over expression of trans-dominant gain of function IKK also (CamK2a-tTA x luciferase-(tetO) IKK2-CA, called IKK2nCA) leads to defects within the dentate gyrus with reduced BDNF expression and higher latency in Morris water mice in 9 month old mice.”

In the section “NF-κB in development” authors mention that “RelA knockout resulted in liver failure due to TNF-alpha-mediated apoptosis”, which to my view is part of the general role for NF-κB as inhibitor of apoptosis.  

In the section “components of NF-κB signaling affected by mutations in humans” authors include the information on epidermal cells “experimentally transfected with the IκB-α super-repressor construct and say “in normal human epidermal cells growth arrest triggered by oncogenic RAS could be bypassed by inhibition of NF-κB with IκB-α thus generating malignant human epidermal tissue”. I do not think that this refers to mutations found in humans.

They also say “Surprisingly, male individuals and male mice with Nemo deficiency die by TNF-mediated liver apoptosis, while females develop normally”. I do not find this observation so surprising, being Nemo located at chromosome X (which is not mentioned).

In addition, there is no references to the detection and possible effects of IκB-α loss (or inactivation) in different human malignancies such as Hodgkin’s lymphoma or glioblastoma. But in contrast, the section includes information relative to manipulation of NF-κB elements in cell lines (I understand that humans refer to persons, not cell lines). For example, “CRISPR/Cas9 mediated knockout of both RELA alleles in human HeLa cells resulted in a profound defect in cell cycle”, “REL-/- cells were more resistant to cisplatin. In human kidney cells HEK293, double knockout of IKK1 and IKK2 blunts NF-κB activation and renders cells to 10 times more sensitive to TNF-induced cell death” and then “We conclude that defects of REL in human cells were profound in cell cycle, which might explain why no human vital mutant with cREL defects was observed so far“, when the title of this section is “NF-κB signaling affected by mutations in humans”.

Thus, to my view this manuscript does not fulfill its function, which should be  that illustrating non- NF-κB experts on the basics of this signaling pathway, its evolution and contribution to human disease.

Author Response

  1. This review focuses on the evolution of NF-κB signaling and its clinical implications. However, the review is not balanced since some sections describing the developmental defects associated with NF-κB mutations are clear and informative while others and misleading. In addition, the more general sections on NF-κB regulation, IκBs and IKKs are highly confusing and sometimes repetitive.

The authors wish to thank the referee for his positive comments on the developmental defects and NF-κB mutations. Now we have improved our text by using the language editing service by MDPI (specialist edit life science). In addition, we have rewritten the more general sections on NF-κB regulation according to the suggestions of the referee.

  1. Multiple sentences and concepts in the review are obscure and the messages incomplete or biased, when the objectives of a review manuscript are to provide useful and understandable information to the readers. Only some examples:

“In this review, we discuss a role of NF-κB in development from Drosophila via Xenopus and mice to humans.” What does it mean “via xenopus and mice”? Are authors referring to a specific evolutionary axis? Then, they also talk about fish, is fish in the same axis?

We thank the reviewer for raising this helpful question. We have tried to introduce the concept of evolutionary developmental biology (evo-devo, Nature Reviews Genetics volume 8, pages 943–949 (2007)) in our review, which we now try to make more clear with additional explanation: Evo-devo (evolutionary developmental biology) is a concept referring to developmental cues in the evolution of species. Here, we used Drosophila as a model of insecta in comparison to higher developed animals such as vertebratae: frog, mouse and men. We used evo-devo to pin point universal concepts of NF-κB signaling (or its homologues) conserved during evolution. We now specify this concept with the abstract and the main text.

  1. “NF-κB is formed by a heteromeric DNA-binding dimer e.g. RelA (p65) and NFKB1 (p50) complexed with IκB”. I do not think that IκB is a subunit of the NF-κB complex. This could apply, at most, to the cytoplastic (inhibited) complex.

In our humble opinion, latent NF-κB is formed by DNA-subunits together with IκB. For reference please see below: “In unstimulated cells, such as “resting” lymphocytes before antigen encounter, NF-κBs are mainly cytoplasmic due to the binding of a dedicated set of inhibitory proteins comprising the “Inhibitor of κB” (IκB) family (Hayden and Ghosh, 2008).” Taken from Zhang et al., 2017. We now added clarifying sentences in the main text and figure legend of figure 1.

  1. Extracellular stimuli of NF-κB include bacterial and viral products, inflammatory cytokines, reactive oxygen, ultra- 40 violet light and even neurotransmitters such as glutamate [8,9]. They activate a kinase complex (IKK, IκB kinase, see Fig. 1) specific for IκB”. Saying that IKK complex is specific for IκB is a statement that shouldn’t be done since several non-IkB targets for IKKs have been identified including FOXO3A, SMRT, SNAP-23, histone H3 or ATM.

We thank the reviewer for this remark. We would like to say that IKK is specific for IκB target, but according to David Baltimore (Cell. 2017 January 12; 168(1-2): 37–57. doi:10.1016/j.cell.2016.12.012), “a kinase complex called IκB kinase (IKK) specifically phosphorylates IκB proteins leading to their degradation” during canonical NF-κB-signaling, which is in accordance to a review by Ben-Neriah from 2002 (Ben-Neriah, 2002, Nat Immunol. 2002; 3:20–26). We now included these references in the respective sentence to better guide the reader.

  1. “IKK complex catalyzes a sequence-specific phosphorylation (serine 32, 36) and ubiquitination of the heterodimer-associated IκB”. They are referring to IκBalpha since residues phosphorylated in other IκBs are different.

We thank the reviewer for this correction. We now revised “IκB” to “IκBα” in the respective sentence and the following ones.

  1. “Developmental pathways would suggest a role of frog NF-κB in development.” I do not understand the meaning of the sentence.

Thank you for the advice, we rephrased this sentence for better clarity: “Following the evo-devo concept, developmental pathways in the lower developed fly may also suggest a role of NF-κB in frog development”.

  1. In the section “NF-κB in development” authors mention that “RelA knockout resulted in liver failure due to TNF-alpha-mediated apoptosis”, which to my view is part of the general role for NF-κB as inhibitor of apoptosis.

Thank you for sharing your opinion. From our point of view, the liver phenotype of RelA KO mice could be rescued by a double KO of TNFR1 and RELA (see Nature Neuroscience volume 6, pages 1072–1078 (2003). Thus, we conclude that TNFα is a major driver of liver apoptosis is RELA-KO mice. We now included this explanation in the main text.

  1. “over expression of trans-dominant gain of function IKK also (CamK2a-tTA x luciferase-(tetO) IKK2-CA, called IKK2nCA) leads to defects within the dentate gyrus with reduced BDNF expression and higher latency in Morris water mice in 9 month old mice.”

Thank you for this remark, but we did not get your point.

  1. In the section “NF-κB in development” authors mention that “RelA knockout resulted in liver failure due to TNF-alpha-mediated apoptosis”, which to my view is part of the general role for NF-κB as inhibitor of apoptosis. 

See response to comment 7.

  1. In the section “components of NF-κB signaling affected by mutations in humans” authors include the information on epidermal cells “experimentally transfected with the IκB-α super-repressor construct and say “in normal human epidermal cells growth arrest triggered by oncogenic RAS could be bypassed by inhibition of NF-κB with IκB-α thus generating malignant human epidermal tissue”. I do not think that this refers to mutations found in humans.

Thank you for this remark. We agree with the reviewer and have deleted this sentence.

  1. They also say “Surprisingly, male individuals and male mice with Nemo deficiency die by TNF-mediated liver apoptosis, while females develop normally”. I do not find this observation so surprising, being Nemo located at chromosome X (which is not mentioned).

Thank you for this helpful comment. We now include this information in the main text and revised the sentence accordingly.

  1. In addition, there is no references to the detection and possible effects of IκB-α loss (or inactivation) in different human malignancies such as Hodgkin’s lymphoma or glioblastoma. But in contrast, the section includes information relative to manipulation of NF-κB elements in cell lines (I understand that humans refer to persons, not cell lines). For example, “CRISPR/Cas9 mediated knockout of both RELA alleles in human HeLa cells resulted in a profound defect in cell cycle”, “REL-/- cells were more resistant to cisplatin. In human kidney cells HEK293, double knockout of IKK1 and IKK2 blunts NF-κB activation and renders cells to 10 times more sensitive to TNF-induced cell death” and then “We conclude that defects of REL in human cells were profound in cell cycle, which might explain why no human vital mutant with cREL defects was observed so far“, when the title of this section is “NF-κB signaling affected by mutations in humans”.

Thank you for this constructive remark. We now revised the subtitle and included relevant citation for morbus Hodgkin and glioblastoma.

  1. Thus, to my view this manuscript does not fulfill its function, which should be  that illustrating non- NF-κB experts on the basics of this signaling pathway, its evolution and contribution to human disease.

We made every effort to address these comments more precisely by professional copy editing and revising every point raised by the referee.

Reviewer 2 Report

In this review article by Kaltschmidt et al, the authors discussed the role of the NF-kappaB pathway in stem cell development and function. This is a well-written report starting from introduction of the canonical NF-kappaB signaling, to the involvement in development of various types of stem cells in terms of differentiation potential. However, there are a number of minor concerns that need to be addressed before this manuscript is in a publishable fashion. Specific comments are as follows:

1. It is suggested that the non-canonical NF-kappaB signaling should also be introduced as RelB, p52 and p100 were also described in later sections.

2. Integrating the part that describes the effect of miRNA on embryonic stem cell development into one section or paragraph is recommended.

3. This article needs a conclusion/future directions section, especially for delivering an overall idea of the NF-kappaB signaling in differentiation of various types of stem cells.

Author Response

In this review article by Kaltschmidt et al, the authors discussed the role of the NF-kappaB pathway in stem cell development and function. This is a well-written report starting from introduction of the canonical NF-kappaB signaling, to the involvement in development of various types of stem cells in terms of differentiation potential. However, there are a number of minor concerns that need to be addressed before this manuscript is in a publishable fashion. Specific comments are as follows:

  1. It is suggested that the non-canonical NF-kappaB signaling should also be introduced as RelB, p52 and p100 were also described in later sections.

We thank the reviewer for this improving remark. We now include an introduction regarding non-canonical NF-kappaB signaling in the first chapter of our review as well as within figure 1.

  1. Integrating the part that describes the effect of miRNA on embryonic stem cell development into one section or paragraph is recommended.

We thank the reviewer for this helpful comment. We now separated the discussion regarding the effects of miRNAs into a distinct paragraph within chapter 4 and refer to this part of the chapter after introducing miR 290-dependent repression of NF-κB translation in ESCs.

  1. This article needs a conclusion/future directions section, especially for delivering an overall idea of the NF-kappaB signaling in differentiation of various types of stem cells.

We thank the reviewer for this constructive remark. We now added a conclusion section particularly summarizing NF-κB signaling in differentiation of stem cells.

Reviewer 3 Report

The manuscript of Kaltschmidt et al. is a comprehensive review on NFKB family of transcription factors and its role in stem cells and development. Although the paper mainly describes most of the aspects related to NFKB biology, there are still some issues that need to be addressed before the manuscript should be accepted for publication:

  • the Authors should revise the text thoroughly and correct the misspellings and grammar mistakes.
    Line 35 : For nuclear import, a disordered NLS sequence ARE important
    Line 72: Nobel Price
    Line 107: Xenopus leavis
    etc.
  • Authors should improve the figures in the manuscript with more original and more complex schemes. For instance, fig. 1 only shows NFKB interaction with iKB and its translocation in the nucleus. Authors should invest more in originality.
  • what is the novelty of this review? the Authors should point out the novelty elements or approach in this review in the conclusions of this paper.
  • the Authors are strongly advised to introduce a Discussion section where to synthesize the message of this review. For the level of "Cells" journal, the review should have a more complex structure, to include Discussions and Conclusions sections, not only an overview of the literature on the subject.
  • in my opinion, the section presenting NFKB role in differentiation and migration of adult mesenchymal stem cells is too briefly described. Considering the adult mesenchymal stem cells subtypes and the multiple types of differentiation they can support, it is insufficient to address only NFKB role in osteogenic differentiation.

Author Response

The manuscript of Kaltschmidt et al. is a comprehensive review on NFKB family of transcription factors and its role in stem cells and development. Although the paper mainly describes most of the aspects related to NFKB biology, there are still some issues that need to be addressed before the manuscript should be accepted for publication:

  • the Authors should revise the text thoroughly and correct the misspellings and grammar mistakes.
    Line 35 : For nuclear import, a disordered NLS sequence ARE important
    Line 72: Nobel Price
    Line 107: Xenopus leavis
    etc.

We thank the reviewer for this helpful advice. We now corrected the respective misspellings and grammar mistakes and further improved our text by using the language editing service by MDPI (specialist edit life science).

  • Authors should improve the figures in the manuscript with more original and more complex schemes. For instance, fig. 1 only shows NFKB interaction with iKB and its translocation in the nucleus. Authors should invest more in originality.

We thank the reviewer for this improving remark, we now revised figure 1 and extended the figure towards non-canonical NF-κB-signaling as well as distinct receptors initiating canonical and non-canonical NF-κB-activity.

  • what is the novelty of this review? the Authors should point out the novelty elements or approach in this review in the conclusions of this paper.
  • the Authors are strongly advised to introduce a Discussion section where to synthesize the message of this review. For the level of "Cells" journal, the review should have a more complex structure, to include Discussions and Conclusions sections, not only an overview of the literature on the subject.

We thank the reviewer for this constructive remark. We now added a conclusion section particularly summarizing the novel key discussion elements of this review and the overall approach.

  • in my opinion, the section presenting NFKB role in differentiation and migration of adult mesenchymal stem cells is too briefly described. Considering the adult mesenchymal stem cells subtypes and the multiple types of differentiation they can support, it is insufficient to address only NFKB role in osteogenic differentiation.

We thank  the reviewer raising this helpful remark. We now extended our discussion towards the role of NF-κB in chondrogenic and myogenic differentiation of MSCs.

Round 2

Reviewer 1 Report

The review is now considerably improved. However, I have identified few mistakes and recommendations:

“Evoluation” in line 108

In lines 135-137: “… we could show that TGFbeta is essential for…“ This needs the reference.

The sentence in lines 196-198 is still difficult to understand. Consider “high latency in the “Morris Water Maze” behavioral test in nine-month-old mice”.

In the section describing alteration in NF-kB elements in human disease, NKBIA deletions in glioblastoma are not even mentioned (Bredel et al, NEJM 2011 and other reports). To my view, this is important.

In line 449, “In summary, the present review summarizes…” is redundant

Finally, if authors want to emphasize de concept on NF-kB conservation during evolution, some data on hydra need to be included and discussed, as well as the fact that NF-kB factors are absent from nematodes that, unexpectedly, do contain IkB-like homologs.

Author Response

The review is now considerably improved. However, I have identified few mistakes and recommendations:

“Evoluation” in line 108

 We thank the reviewer for pointing out this misspelling, which we now corrected.

In lines 135-137: “… we could show that TGFbeta is essential for…“ This needs the reference.

We thank the referee for this helpful comment. We now included the respective reference “TGF-β2 neutralization inhibits proliferation and activates apoptosis of cerebellar granule cell precurors in the developing cerebellum”, Mechanisms of Development, Volume 122, Issue 4, April 2005, Pages 587-602 by M. Elvers, J. Pfeiffer, C. Kaltschmidt and B. Kaltschmidt.

The sentence in lines 196-198 is still difficult to understand. Consider “high latency in the “Morris Water Maze” behavioral test in nine-month-old mice”.

We thank the referee for this improving correction. We now rephrased the sentence accordingly.

In the section describing alteration in NF-kB elements in human disease, NKBIA deletions in glioblastoma are not even mentioned (Bredel et al, NEJM 2011 and other reports). To my view, this is important.

We thank the referee for this important comment. We now include the study by Bredel regarding NKBIA deletions in glioblastoma within section 3 (see lines 230-238 in the revised manuscript).

In line 449, “In summary, the present review summarizes…” is redundant

We now rephrased the sentence accordingly.

Finally, if authors want to emphasize de concept on NF-kB conservation during evolution, some data on hydra need to be included and discussed, as well as the fact that NF-kB factors are absent from nematodes that, unexpectedly, do contain IkB-like homologs.

We thank the referee for raising this improving comment. We now discuss the presence of NF-κB homologs in Cnidaria like the sea anemone Nematostella vectensis or the hydra Hydractinia symbio-longicarpus. In addition, we address the contradiction of IκB homologs like NFKI-1 and IKB-1 being present in nematodes and vital for their development, although other NF-κB-homologous proteins are absent in nematodes (see lines 102-111 in the revised manuscript).

Reviewer 3 Report

The Authors have significantly improved the quality of the manuscript by adding important knowledge on NFKB involvement in MSC differentiation and in other biological processes. Most importantly, they have introduced a new section summarizing the main ideas of the review which they named "Conclusions". In my opinion, considering its length and complexity, it would be more appropriate to name it "Discussion and concluding remarks". Otherwise, the manuscript is now improved and suitable for publication in my opinion.

Author Response

The Authors have significantly improved the quality of the manuscript by adding important knowledge on NFKB involvement in MSC differentiation and in other biological processes. Most importantly, they have introduced a new section summarizing the main ideas of the review which they named "Conclusions". In my opinion, considering its length and complexity, it would be more appropriate to name it "Discussion and concluding remarks". Otherwise, the manuscript is now improved and suitable for publication in my opinion.

We thank the referee for raising this helpful comment. We now renamed the section towards "Discussion and concluding remarks”.